

# Global changes of miRNA expression indicates an increased reprogramming efficiency of induced mammary epithelial cells by repression of miR-222-3p in fibroblasts

Mingxing Liu[1,2], Siyi Liu[2], Liangshan Qin[2], Danwei Lv[2], Guodong Wang[2], Quanhui Liu[2], Ben Huang[1,2] and Dandan Zhang[1]

[1] Guangxi Key Laboratory of Eye Health, Guangxi Academy of Medical Sciences, The People's Hospital of Guangxi Zhuang Autonomous Region, Nanning, Guangxi, China
[2] Guangxi University, School of Animal Science and Technology, Nanning, Guangxi, China

Corresponding authors
Ben Huang, bhuang@gxams.org.cn
Dandan Zhang, ddzhang@gxams.org.cn

## ABSTRACT

**Background:** Our previous studies have successfully reported the reprogramming of fibroblasts into induced mammary epithelial cells (iMECs). However, the regulatory relationships and functional roles of MicroRNAs (miRNAs) in the progression of fibroblasts achieving the cell fate of iMECs are insufficiently understood.

**Methods:** First, we performed pre-and post-induction miRNAs sequencing analysis by using high-throughput sequencing. Following that, Gene Ontology (GO) and Kyoto Encyclopedia of Genes and Genomes (KEGG) pathway enrichment studies were used to determine the primary roles of the significantly distinct miRNAs and targeted genes. Finally, the effect of miR-222-3p on iMECs fate reprogramming *in vitro* by transfecting.

**Results:** As a result goat ear fibroblasts (GEFs) reprogramming into iMECs activates a regulatory program, involving 79 differentially expressed miRNAs. Besides, the programming process involved changes in multiple signaling pathways such as adherens junction, TGF-β signaling pathway, GnRH secretion and the prolactin signaling pathway, *etc*. Furthermore, it was discovered that the expression of miR-222-3p downregulation by miR-222-3p inhibitor significantly increase the reprogramming efficiency and promoted lipid accumulation of iMECs.

## INTRODUCTION

Downstream developmental lineages of cells derived from changes in the initiating cell type are regulated by transcription factors (*Yu et al., 2014*). Therefore, establishing a new balance of the transcriptional network is essential for a cell to move toward a new fate. Previous research has established that somatic cells can be directly reprogrammed into hepatocytes (*Du et al., 2014*) neurons (*Vierbuchen et al., 2010*) and cardiomyocyte lineages by lineage-specific transcription factors (*Ieda et al., 2010*). A growing body of literature has

shown that small molecule compounds promote transcription factor-based reprogramming by modulating signaling pathways (*Qin, Zhao & Fu, 2017*).

As small noncoding RNA, miRNAs can target many mRNAs and play a role in posttranscriptional gene regulation through protein interactions in the RNA-induced silencing complex (*Selbach et al., 2008*). It has been demonstrated that miRNAs are essential to somatic cell reprogramming, allowing mature double-stranded miRNAs to be directly transfected into mouse and human cells to reprogramme them to become pluripotent (*Miyoshi et al., 2011*). It is now well established from a variety of studies that miRNAs are key regulators of mammary epithelial cells (MECs) at different developmental stages (*Jena, 2017*). MiRNAs can affect mammary development by affecting hormone level expression (*Chen et al., 2018*). In conclusion, miRNAs are essential for reprogramming and mammary gland development.

MECs have been the only target cell type needed for recombinant protein production in mammary bioreactors due to their unique ability to lactate (*Ma et al., 2017*). MECs can be obtained from the mammary epithelium by collagenase digestion or directly from the breast milk of different species (*Lasfargues, 1957*; *Boutinaud, Herve & Lollivier, 2015*). Such approaches, however, have failed to address MECs obtained had too low for transgenic protein expression in milk-producing mammary bioreactors.

We successfully reprogrammed goat ear fibroblasts (GEFs) into iMECs using five small-molecule compounds (*Zhang et al., 2021*). Previous studies of the mechanism of this reprogramming process have not addressed the miRNA levels. In the present study, we analyzed small RNA sequencing before and after iMEC induction obtained from a combination of five small molecule compounds. Furthermore, we validated the effect of miR-222-3p on the transition of goat ear fibroblasts to a mammary epithelial cell fate identity.

## MATERIALS AND METHODS

### Sample collection

Goat ear fibroblasts were preserved by the College of Animal Science and Technology, Guangxi University. Based on the former research results, goat ear fibroblasts were seeded into a 60-mm cell culture dish at a density of $5 \times 10^5$ cells per dish. Following 24 h, the medium was changed to "VTFBR" (500 μg/mL VPA, 10 μM Tranylcypromine, 10 μM Forskolin, 1 μM TTNPB, 10 μM RepSox), a cocktail induction medium based on N2B27 and small molecule chemicals (*Zhang et al., 2021*). The induction culture was maintained for 8 days with medium changes every 2 days.

### Creating small RNA libraries

Total RNA was isolated from reprogrammed iMECs cells using TRIzol reagent at the predetermined time periods (the initiating GEFs (0 day) and the day 8 post-induction iMECs), applying three replicates for each time point, and then pooled together for sequencing in a total of six libraries. A total of 1% gel electrophoresis was employed to detect total RNA whether degradation. Using 8% polyacrylamide gel electrophoresis, whole RNA was separated into fragments between 15 and 35 nt, and then the fragments

were linked with proprietary adapters. Reverse transcription was used to create complementary DNA (cDNA) in order to create the libraries for sequencing. Using Illumina HiSeq to perform RNA sequencing with the SE50 sequencing strategy (*Hafner et al., 2008*).

## Data analysis

Clean reads for subsequent analysis were obtained from raw bases corresponding to the sequenced reads by removing joints, removing low quality, and selecting fragments (*Lindgreen, 2012*). Clean reads in the genome were matched with miRNAs localization information (100% overlap) (*Quinlan & Hall, 2010*). Using miRDeep2 predicted novel miRNAs whose sequences have been discovered. Normalization was performed for each sample to obtain the trimmed mean of M-values (TMM). DESeq2 was used to identify differentially expressed miRNAs. When $|\log_2 FC| > 1$, $p$adj < 0.05 and Cohen's d > 1, the miRNAs were considered as differentially expressed (*Quinlan & Hall, 2010*).

## MiRNAs enrichment analysis

GO and KEGG analyses of differentially expressed miRNAs were performed using the miRPath V4.0 database (*Tastsoglou et al., 2023*).

## Cell transfection and induction

Goat ear fibroblasts were inoculated into 6-hole plate at a density of $2 \times 10^5$ cells per dish, and cultured in Dulbecco's Modified Eagle's Medium supplemented with 10% (v/v) fetal bovine serum at 37.5 °C in a humidified atmosphere of 5% $CO_2$ and 95% air. Cell transfection was performed when cell confluence reached 60%. The ransfected cells were divided into miR-222-3p mimic group, miR-222-3p mimic-NC group, miR-222-3p inhibitor group, and miR-222-3p inhibitor-NC group. Afterwards, lipofectamine TMRNAiMAX and each group of miRNA fragments were transfected into cells as described in the manufacturer's protocol, and three biological replicates of each group were performed. After 12 h, the medium was replaced with N2B27 and small molecule compounds cocktail induction medium "VTFBR" (500 µg/mL VPA, 10 µM tranylcypromine, 10 µM Forskolin, 1 µM TTNPB, 10 µM RepSox) at 37.5 °C in a humidified atmosphere of 5% $CO_2$ and 95% air. The medium was changed every 2 days, and the induction culture was continued for 4 days. Three biological replicates were performed for each group. Cell induction was observed under a microscope every 2 days and photographed.

## Oil red O staining and quantification

Four groups of iMECs were transfected for 4 days, and the medium in the petri dishes was aspirated medium. Cells were subsequently fixed with 4% paraformaldehyde for 10 min and then stained with oil red O staining solution for 15 min. The cells were then rinsed with 60% isopropanol for 15–30 s, washed three more times with PBS, and observed under a microscope. Finally for quantitative analysis of lipid accumulation, lipids were washed off using 100% isopropanol and then absorbance at 510 nm was measured using a spectrophotometer. Three biological replicates of each group were performed.

## Immunofluorescence staining

The cells were permeabilised with 1% Triton X-100 after fixation by 4% paraformaldehyde for half an hour at room temperature. Subsequently, they were closed with 10% donkey serum for 1.5 h. Primary antibodies were incubated with the cells overnight at 4 °C at the appropriate dilution in containment buffer. The next day, the cells were incubated with the respective secondary antibodies for 1 h in the dark at room temperature (*Zhang et al., 2021*). Photographic documentation using a microscope. Quantification of iMEC using ImageJ software. Three biological replicates were performed for each group.

## Statistical analysis

Reprogramming efficiency, the quantitation of Oil Red O staining and immunofluorescence were carried out by one-way ANOVA in Graphpad Prism 8.0 analyzer. Data are presented as mean ± SEM of independent biological replicates. *$p < 0.05$, **$p < 0.01$, ***$p < 0.001$. The experimental data all have three biologically independent samples.

# RESULTS

## Introduction to sequencing data

Based on our past experimental results, six small RNA-seq libraries were constructed for high-throughput sequencing, with three biological repetitions at each time-point (GEFs and iMECs) (*Zhang et al., 2021*). For the six small RNA libraries, the averages of 13,372,156.33 raw data were obtained, and the averages of 11,180,154.50 clean reads were obtained (Table S1). Lengths were ranged from 18 to 30 nt, and the peak appeared at 22 nt which conformed to the law of animal small RNA (Fig. 1A). The Q20 values of the miRNA libraries were all higher than 97%. The length distribution of small RNAs in GEFs and iMECs follows the typical read distribution of animal miRNAs, mostly in the range of 22 nt. The clean reads were contrasted to the goat reference sequence (Table S2). Principal component analysis (PCA) (Fig. 1B) was run to cluster the samples based on the expression values and showed a significant difference before and after induction between GEFs and iMECs, the main difference lay in pc1. We also showed the density distribution to visually demonstrate the expression level of all the genes that were detected in the two samples (Fig. 1C). In conclusion, the miRNA libraries used in the current study had high enough data quality to be used for subsequent statistical calculations.

## Differentially expressed miRNA and GO and KEGG enrichment analysis

All six miRNA libraries detected a total of 457 miRNAs, including 116 novel miRNAs and 341 known miRNAs. The miRNAs showed were shown differential expression patterns (Fig. 2A). Compared with the GEF libraries, 33 known miRNAs were upregulated in the iMECs, and 46 known miRNAs were downregulated with $|\log_2 FC| > 1$, $-\log10(p$ value$) <$ 0.05 and Cohen's d > 1 (Figs. 2B, 2C, Table S3). From the graph above we can see that, chi-miR-34c-3p, chi-miR-34b-3p, chi-miR-34b-5p, chi-miR-34c-5p, and chi-miR-141 were the most obviously upregulated, but chi-miR-100-3p, chi-miR-296-3p, chi-miR-155-3p,
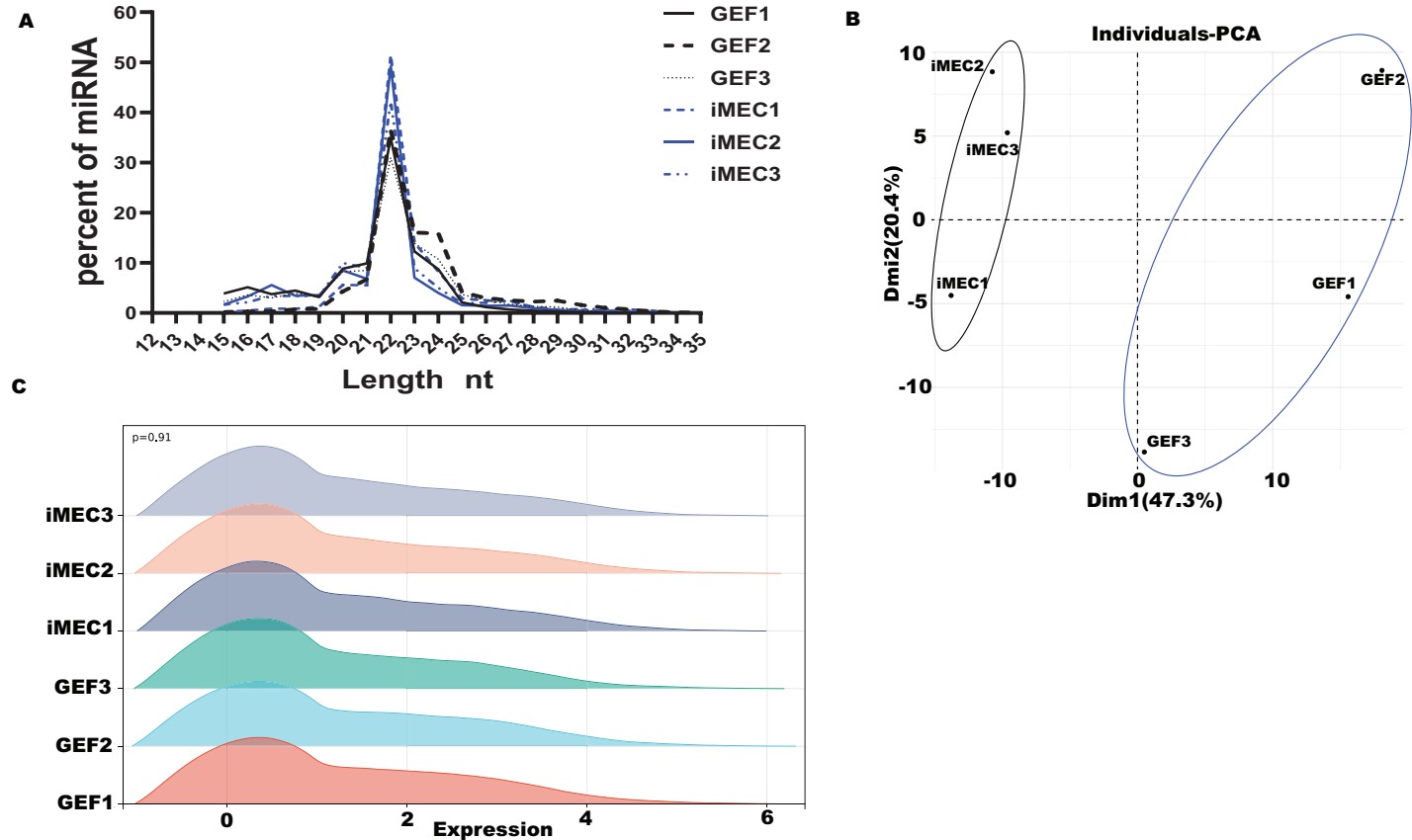

**Figure 1 Quality control.** (A) The size distribution of the small RNAs found in the mammary gland between GEFs groups and iMECs groups. (B) PCA of gene expression values (FPKM) for the three sets of samples. (C) The gene expression levels of each sample in the GEFs groups and iMECs groups were demonstrated.

chi-miR-665, and chi-miR-487b-3p were the most significantly downregulated. This suggested that these miRNAs may play a key role during mammary gland lineage formation. Interestingly, the expression of all 34 family members was observed to be significantly elevated in iMECs.

Both GO and KEGG analyses were carried out to mine cellular biological processes and signaling pathways, respectively, in order to highlight the relevance of differentially expressed miRNAs in the reprogramming of GEFs to iMECs. The KEGG pathway analysis results indicated the involvement of these DE-miRNAs mostly in adherens junction, TGF-β signaling pathway, GnRH secretion and the prolactin signaling pathway, *etc*. (Fig. 2D). According to the significance level of GO term enrichment, cells before and after induction were mainly enriched in the molecular functional classifications of protein interactions, enzyme activities, transcriptional regulation, as well as classification of cellular components of cell structure, organelles, protein complexes and cellular transport. In addition, biological processes involved cell cycle, cellular response to DNA damage stimuli and protein phosphorylation (Fig. 2E). Furthermore, we observed that downregulated miRNAs were associated with some hormone pathways (Table S4) while upregulated miRNAs were associated with some EMT pathways (Table S5). An explanation for this

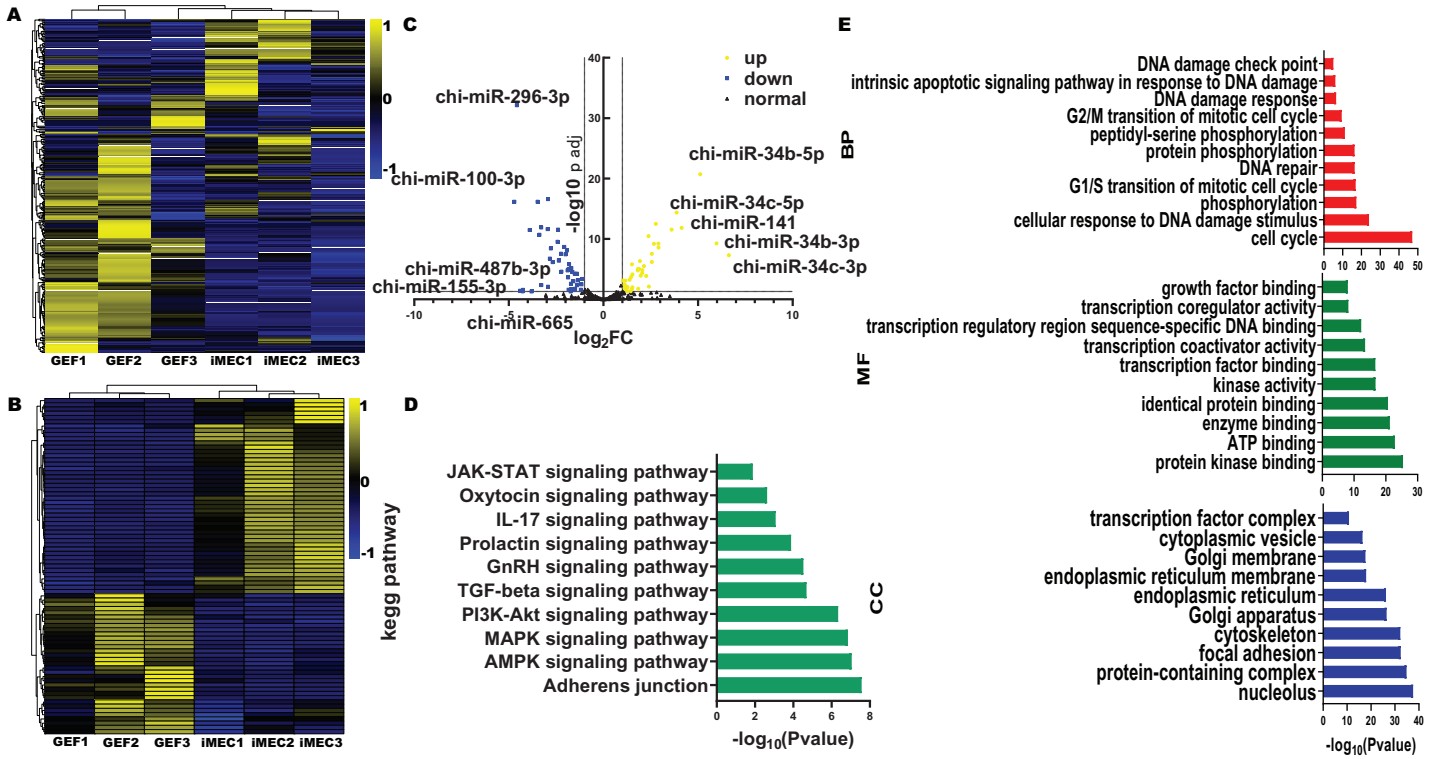

**Figure 2 Differentially expressed miRNA And GO and KEGG enrichment analysis.** (A) Clustering heatmap of all miRNAs. The expression levels of miRNAs range from high to low, indicated by the color gradient changing from yellow to blue. (B) Clustering heatmap of 79 differentially expressed miRNAs. The expression of miRNAs from high to low is indicated by the change from yellow to blue. (C) The differential gene volcano map between groups GEF and iMEC (top five up-regulated and down-regulated miRNAs are labeled). (D) The top 10 significantly enriched KEGG pathways of differentially expressed miRNAs among the GEFs groups and the iMECs groups. (E) The top 10 significantly enriched GO annotation of differentially expressed miRNAs among the GEFs groups and the iMECs groups.

might be that upregulated miRNAs maintained epithelial cell characteristics by suppressing EMT, and downregulated miRNAs gave hormonal signals that allowed them to become a mammary gland spectrum.

## MiR-221-3p plays a pivotal role in the generation of induced mammary epithelial cells, thus highlighting its significant contribution to this process

We found that among the differentially expressed miRNAs, miR-222-3p was significantly higher than other miRNAs expressed in fibroblasts in sequencing data. Meanwhile miR-222-3P not only regulates mesenchymal-epithelial transition but also plays an important role in adipogenesis. Therefore to investigate the impact of miRNAs on reprogramming efficiency of induced mammary epithelial cells, we focused on miR-221-3p as our target. Prior to induction with five small molecule compounds (VTFBR), goat ear fibroblasts were transfected with either a miR-221-3p mimic or inhibitor to upregulate or downregulate miR-221-3p expression, respectively. Our results demonstrated that cultures treated with the miR-221-3p inhibitor exhibited an increased number of independent and compact epithelial cell-like colonies, whereas fewer colonies were observed in cultures transfected

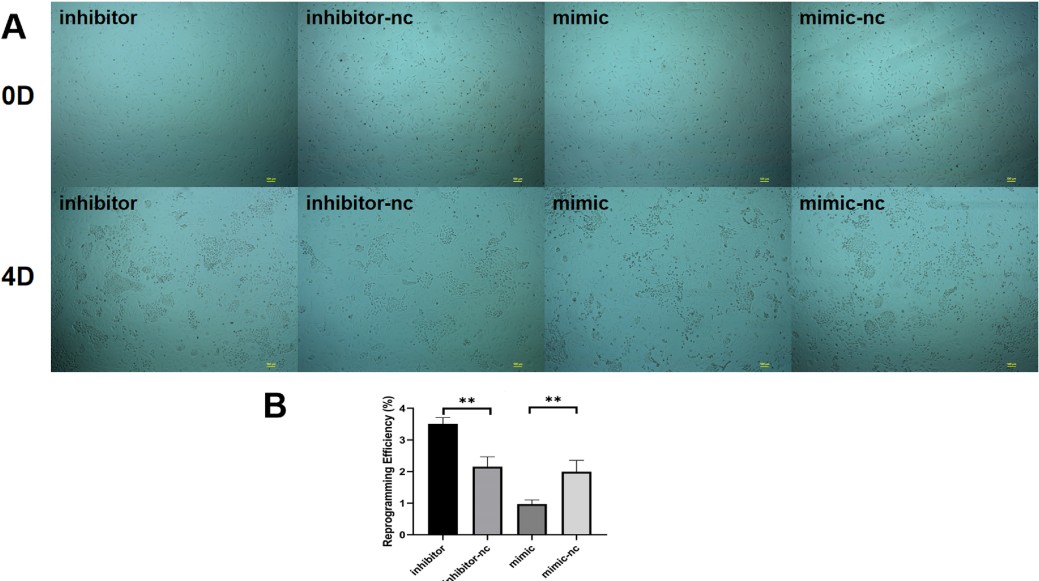

**Figure 3 Effect of miR-221-3p on reprogramming efficiency of goat induced mammary epithelial cells.** (A) The ability of cells to form independent clones within 4 days under transfection miR-221-3p inhibitor, miR-221-3p inhibitor-NC, miR-221-3p mimic, miR-221-3p mimic-NC. Scale bar, 100 μm. (B) Reprogramming efficiency of goat induced mammary epithelial cells under transfection (Efficiency (%) = No. of primary colonies/No. of seeded cells × 100%). Scale bar, 100 μm. $n$ = 3 biological replicates. Data are represented as the mean ± SEM. **$p < 0.01$, ***$p < 0.001$ (one-way ANOVA).

with the miR-221-3p mimic compared to controls at 4 days (Fig. 3A). These findings suggest that upregulation of miR-221-3p by the mimic severely impairs small-molecule compound-induced colony formation and significantly reduces reprogramming efficiency (Fig. 3B). Thus, our observations indicate that elevated expression of miR-221-3p may hinder fibroblast-to-iMEC transformation.

## MiR-221-3p mediates the biological properties of induced mammary epithelial cells

Subsequently, to gain deeper insights into the impact of miR-222-3p on iMEC functionality, we assessed the expression levels of MEC specific proteins and milk fat synthesis. The secretion of milk lipids represents a crucial characteristic of iMECs, and our findings revealed that treatment with an inhibitor targeting miR-221-3p resulted in enhanced cytoplasmic lipid droplet accumulation as evidenced by saturated oil red O staining. Conversely, iMECs treated with a mimic for miR-222-3p exhibited significantly reduced secretion of periplasmic lipid droplets (Figs. 4A, 4B). Furthermore, immunofluorescence staining demonstrated elevated expression levels of mammary epithelial cell-specific antigens CDH1, CK8, and CK14 in iMECs treated with the miR-221-3p inhibitor (Figs. 4C, 4D).

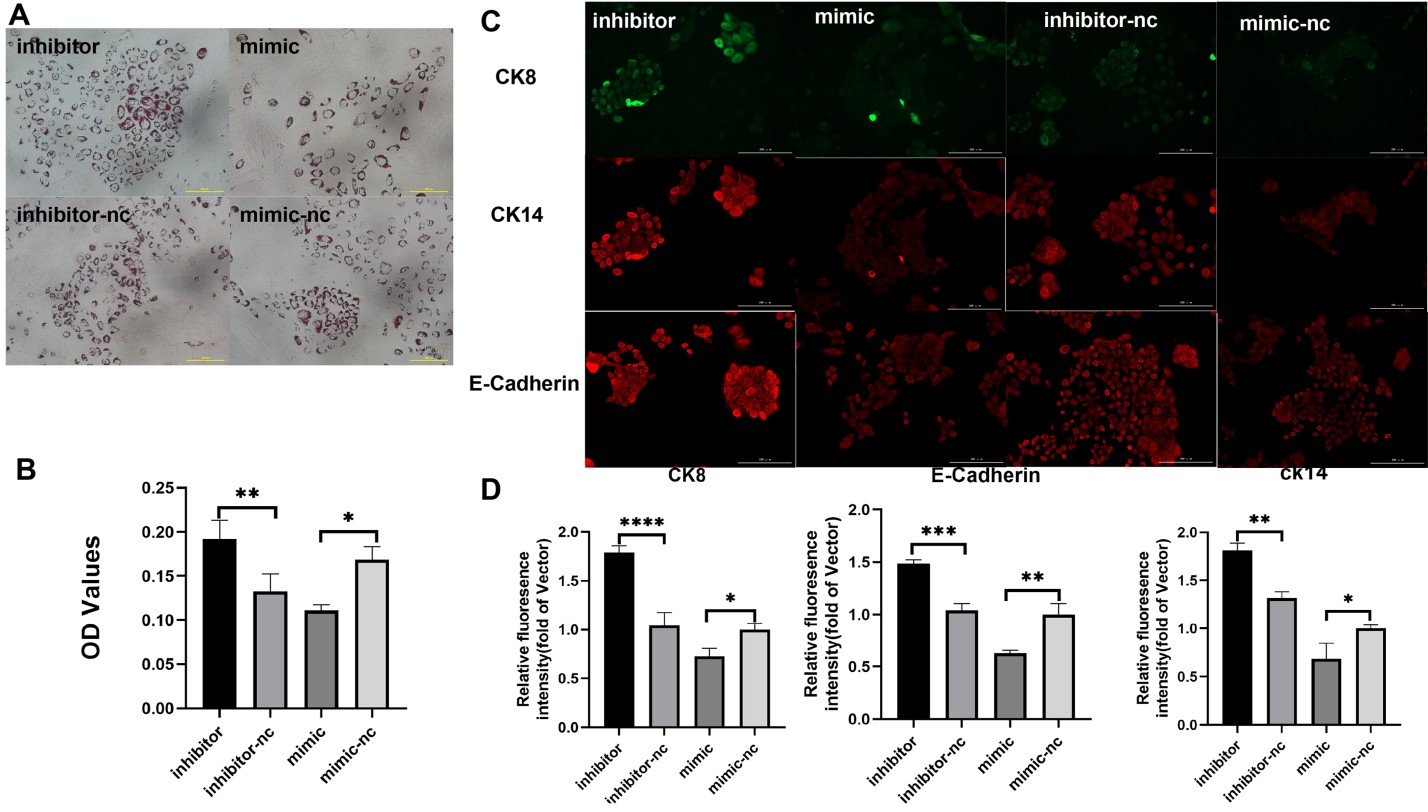

**Figure 4 Biological characterization of iMECs under transfection.** (A) Saturated oil red O staining was performed to identify iMECs under transfection miR-221-3p inhibitor, miR-221-3p inhibitor-NC, miR-221-3p mimic, miR-221-3p mimic-NC. Scale bar, 100 μm. (B) Histogram showing the quantitation of Oil Red O staining by spectrophotometry. $n = 3$ biological replicates. Data are represented as the mean ± SEM. *$p < 0.05$, **$p < 0.01$, ***$p < 0.001$ (one-way ANOVA). (C) Immunofluorescence staining was performed to detect the expression of iMECs under transfection miR-221-3p inhibitor, miR-221-3p inhibitor-NC, miR-221-3p mimic, miR-221-3p mimic-NC. Scale bar, 100 μm. (D) Quantitative analysis of immunofluorescence intensity. Scale bar, 100 μm. $n = 3$ biological replicates. Data are represented as the mean ± SEM. *$p < 0.05$, **$p < 0.01$, ***$p < 0.001$, ****$p < 0.0001$ (one-way ANOVA).

## DISCUSSION

Cell reprogramming techniques allow cells to escape downstream developmental lineages where the initial cell type changes. As we showed in our previous report, iMECs can be chemically reprogrammed from fibroblasts by inhibiting the TGFβR1-Smad3 regulatory site. MiRNAs play an important role in transcriptional regulation and can affect the expression of transcription factors through translational repression or mRNA degradation (*Henzler et al., 2013*). At the same time, miRNAs can break through epigenetic barriers by directly targeting epigenetic regulators such as the histone acetyl transferase p300 as well as the H3K4 demethylase Jarid1a, thereby facilitating reprogramming (*Pfaff et al., 2017*). However, our previous studies did not focus on changes in miRNA levels. We hope to determine how miRNAs maintain the fate of mammary epithelial cells through this study.

In this study, we profiled miRNA expression and screened miRNAs with differential significant expression. We found that the expression of chi-miR-34b-5p, chi-miR-34b-3p, chi-miR-34c-3p, and chi-miR-34c-5p, which are members of the microRNA-34 family, increased most significantly in the differentially expressed miRNAs. This finding was

consistent with that of *Bonetti et al. (2020)*, who discovered that the miR-34 family was essential in the fate commitment and differentiation of mammary epithelial cells. Surprisingly, significantly downregulated mir145, a key target of the tgfB pathway, plays a significant role in the activation of hematopoietic stem/progenitor cells (HSPCs) and fibroblast differentiation (*Lam et al., 2018*; *Fang et al., 2016*). By KEGG analysis, we found that downregulated miRNAs were involved in four hormone-related pathways (GnRH secretion, prolactin signaling pathway, thyroid hormone signaling pathway, VEGF signaling pathway), and upregulated miRNAs were significantly enriched in some EMT-related pathways (focal adhesion, leukocyte transendothelial migration, adherens junction). According to these data, we could infer that miRNAs maintain the fate transformation of MEC lineage cells by activating hormonal signaling and inhibiting EMT signaling. It is worth noting that GSEA of the predicted target genes obtained the same conclusion.

MiRNAs, as important regulators, are involved in almost every step of cellular processes, and although there is a large body of evidence supporting the involvement of miRNAs in the process of cellular reprogramming, little is known about what role miRNAs play in the reprogramming of fibroblasts into mammary epithelial cells. The inhibition of miR-222-3p expression levels was demonstrated by our experiments to be more conducive to fibroblast-to-mammary epithelial cell identity. There is evidence that miR-222-3p affects cancer cell migration by promoting the process of EMT in the development of multiple cancers (*Fuziwara & Kimura, 2014*; *Li et al., 2020*). In our past studies it was thought that fibroblasts were required to undergo MET to achieve the transition to a mammary epithelial cell fate. We therefore speculate that lower expression levels of miR-222-3p may be more helpful for the advancement of the MET process, thereby improving reprogramming efficiency. Next, we experimentally found that miR-222-3p inhibitor transfected cells induced to become imec possessed stronger milk fat secretion ability and expression of mammary epithelial signature proteins. This is consistent with the findings of Pere Bibiloni et al. that miR-222 expression is downregulated in 3T3-L1 cells during adipogenesis (*Xie, Lim & Lodish, 2009*). High expression of CK14, CK8, and E-cadherin proteins demonstrates better maintenance of epithelial cell identity after inhibitor miR222-3P-induced cellular antigens. These evidences may indicate that miR222-3p is one of the barriers for fibroblasts to achieve mammary epithelial cell fate transition, and inhibition of miR222-3p expression facilitates faster and better mammary epithelial cell fate transition.

## CONCLUSIONS

In conclusion, a global view of changes in miRNA levels in GEFs and iMECs before and after reprogramming was obtained by high-throughput sequencing. It was found that differentially expressed miRNAs may drive reprogrammed cells into the mammary developmental lineage by disrupting EMT processes and responses to hormonal signals. More importantly, we verified that inhibition of miR-222-3p improves cloning efficiency and promotes lipid droplet formation for induced mammary epithelial cells. These findings help us to better refine the reprogramming system and accelerate reprogramming

efficiency. However, further search for miR-222-3p downstream target genes and further elucidation of the regulatory mechanism for iMECs still require experimental validation.

The statistical power of this experimental design, calculated in RNASeqPower is 0.7256308.

### Funding

This research supported by the grants from the National Natural Science Foundation of China (Grant No. 32160171), the Natural Science Foundation of Guangxi (Grant No. 2023GXNSFBA026023) and the Specific Research Project of Guangxi for Research Bases and Talents (Grant No. AD23026095). The funders had no role in study design, data collection and analysis, decision to publish, or preparation of the manuscript.

### Grant Disclosures

The following grant information was disclosed by the authors:
National Natural Science Foundation of China: 32160171.
Natural Science Foundation of Guangxi: 2023GXNSFBA026023.
Specific Research Project of Guangxi for Research Bases and Talents: AD23026095.

### Competing Interests

The authors declare that they have no competing interests.

### Author Contributions

- Mingxing Liu conceived and designed the experiments, performed the experiments, prepared figures and/or tables, authored or reviewed drafts of the article, and approved the final draft.
- Siyi Liu performed the experiments, prepared figures and/or tables, and approved the final draft.
- Liangshan Qin analyzed the data, authored or reviewed drafts of the article, and approved the final draft.
- Danwei Lv analyzed the data, prepared figures and/or tables, and approved the final draft.
- Guodong Wang analyzed the data, authored or reviewed drafts of the article, and approved the final draft.
- Quanhui Liu analyzed the data, authored or reviewed drafts of the article, and approved the final draft.
- Ben Huang conceived and designed the experiments, authored or reviewed drafts of the article, and approved the final draft.
- Dandan Zhang conceived and designed the experiments, authored or reviewed drafts of the article, and approved the final draft.

## DNA Deposition

The following information was supplied regarding the deposition of DNA sequences:

The sequencing data (miRNA-seq and mRNA-seq) required to reproduce these findings are available for download from GEO (https://www.ncbi.nlm.nih.gov/geo; Access ID: GSE207893 and GSE142551). URL: https://www.ncbi.nlm.nih.gov/geo/query/acc.cgi?&acc=GSE207893. The security token for database GSE207893 is yjgriquqjlsdzsf.

## Data Availability

The raw data for Figs. 3 and 4 are available in the Supplemental Files.

## Supplemental Information

Supplemental information for this article can be found online at http://dx.doi.org/10.7717/peerj.17657#supplemental-information.

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
