# Peer review of "Global changes of miRNA expression indicates an increased reprogramming efficiency of induced mammary epithelial cells by repression of miR-222-3p in fibroblasts"

_PeerJ, doi:10.7717/peerj.17657_

## Round 0.1 · original submission · Major Revisions

Please address concerns of both reviewers and amend your manuscript accordingly

Reviewer 1 ·

Basic reporting

This paper's got a solid setup, hitting us with some clear goals and what's been done before. But, it kinda trips over its own feet with some parts being tough to get. Plus, the pics and graphs, they're on point but could use a bit more oomph in explaining what's what.

Experimental design

The study's design to explore miRNA's role in cell reprogramming is relevant and meaningful. Yet, the lack of detail regarding experimental controls and the reproducibility of results poses a significant concern. Moreover, the methods section does not provide sufficient detail for replication, such as the specific conditions under which cell transfections were performed.

Validity of the findings

The conclusions drawn from the findings appear overstated given the data presented. The statistical analysis lacks depth, particularly in addressing the biological variability and the robustness of the differential expression results. Additionally, there's a notable absence of external validation for the miRNA targets identified, which weakens the overall argument.

Reviewer 2 ·

Basic reporting

This manuscript lays out an intriguing study on miR-222-3p's role in reprogramming fibroblasts into induced mammary epithelial cells. The creative approach, significant discoveries, and structured experiments offer valuable insights into the processes of miRNA-mediated cellular reprogramming. However, tackling the detailed suggestions, particularly regarding statistical methods and experimental controls, will make the study more rigorous and clear. These adjustments could solidify the conclusions and boost the manuscript's influence in its field.

Experimental design

1. For normalizing the data before delving into the differntial expression analysis, we opted to use Reads Per Million (RPM). Considering the vast and detailed nature of RNA sequencing data, it might be wise to look into more solid normalization tactics that take into account both library size and the various effects of composition, like the Trimmed Mean of M-values (TMM) or Quantile normalization. Doing this could cut down on bias and sharpen the precision of differential expression analysis.
2. In the paper, it's mentioned that miRNAs were tagged as differentially expressed if the |Fold Change| was more than 1 and padj was less than 0.05. But, just sticking to fold change and p-value might miss out on the size of the effect and the variability present within the data. It's recommended to throw in measures of effect size, such as Cohen's d or log odds ratio, to grab a more detailed picture of what's biologically significant about the changes we're observing.
3. The heatmaps (like in Fig 2.A) do a good job at showing the data visually, but adding hierarchical clustering dendrograms for both rows (miRNAs) and columns (samples) could make it even better. This extra step would make it easier for readers to get the hang of how different miRNAs and sample conditions group together and relate to each other, offering insights into the biological importance of these patterns.

Validity of the findings

1. The manuscript would really benefit from a clearer explanation of the hypothesis it's testing, especially about the role of miR-222-3p in turning fibroblasts into mammary epithelial cells. It'd be better to organize the Results section in a way that tackles each part of the hypothesis step-by-step, creating a smooth narrative that leads to the conclusion.
2. The discussion on what happens when miR-222-3p is inhibited is interesting, but the manuscript would be even stronger if it also looked at what happens when miR-222-3p is overexpressed. This would give us a fuller picture of the role miR-222-3p plays in reprogramming and help confirm the direction of its effects.

Additional comments

Although the manuscript does a decent job including relevant keywords, slipping in "Transcriptional Regulation" and "Epigenetic Reprogramming" could make it more accessible. These terms are key because the reprogramming directed by miRNA touches on changes in gene expression and potentially epigenetic shifts, which draws a lot of eyes from folks in developmental biology and regenerative medicine.

---

## Round 0.2 · accepted · Accept

All concerns of the reviewers were adequately addressed and revised manuscript is acceptable now.

Reviewer 1 ·

Basic reporting

none

Experimental design

none

Validity of the findings

none